# Characterization of Flavor Compounds in Distilled Spirits: Developing a Versatile Analytical Method Suitable for Micro-Distilleries

**DOI:** 10.3390/foods11213358

**Published:** 2022-10-25

**Authors:** Quentin Barnes, Jérôme Vial, Didier Thiébaut, Clément De Saint Jores, Damien Steyer, Marie-Anne Contamin, Nicolas Papaiconomou, Xavier Fernandez

**Affiliations:** 1Institute of Chemistry of Nice (ICN), UMR Université Côte d’Azur CNRS 7272, 06108 Nice, France; 2Comte de Grasse, 06130 Grasse, France; 3Department of Analytical, Bioanalytical Sciences and Miniaturization (LSABM), UMR CBI 8231, ESPCI Paris, PSL Research University CNRS, 75231 Paris, France; 4Twistaroma, 67400 Illkirch-Graffenstaden, France

**Keywords:** distilled spirits, micro-distillery, model mixture, aroma compounds, sample preparation, headspace solid phase microextraction (HS-SPME), gas chromatography (GC), mass spectrometry (MS)

## Abstract

Over the last few years, the development of micro-distilleries producing diverse spirits with various flavors has been observed. Versatile analytical techniques for the characterization of aroma compounds in such alcoholic beverages are therefore required. A model mixture embodying a theoretical distilled spirit was made according to the data found in literature to compare usual extraction techniques. When it was applied to the model liquor, the headspace solid phase microextraction (HS-SPME) extraction method was preferred to the liquid-liquid extraction (LLE), solid phase extraction (SPE) and stir bar/headspace sorptive extraction (SBSE/HSSE) methods according to efficiency, cost, and environmental criteria. An optimization study using the model mixture showed that the extraction was optimal with a divinylbenzene/carboxen/poly(dimethylsiloxane) DVB/CAR/PDMS fiber, during 60 min, at 35 °C and with the addition of 10% NaCl. This method was successfully applied to three different commercial liquors and led to the identification of 188 flavor compounds, including alcohols, esters, lactones, carbonyls, acetals, fatty acids, phenols, furans, aromatics, terpenoids, alkenes, and alkanes.

## 1. Introduction

Spirit drinks are defined by the European regulation as beverages intended for human consumption with a minimum alcoholic strength of 15%. They possess particular organoleptic qualities and can be produced using specific methods, including distillation of fermented products and maceration of plant materials in agricultural ethyl alcohol [1].

Flavor compounds in distilled spirits present a large variety of chemical classes, such as esters, alcohols, fatty acids, carbonyls, sulfur compounds, terpenoids, volatile phenols, and heterocyclic compounds. Therefore, they present very diverse aromas. For instance, esters generally have a positive contribution to the flavor profile with fruity notes and heterocycles like furans are responsible for caramellic aromas. They are formed during different stages of the production process [2]. Raw materials containing carbohydrates subjected to alcoholic fermentation are a primary source of aromas. In addition to ethanol and carbon dioxide, the action of yeasts leads to the formation of volatile compounds that participate to the flavor profile. Distillation further enables the concentration of desirable organoleptic compounds as well as the reduction of unpleasant sulfur compounds [3]. Finally, during aging in oak casks, molecules such as whiskey lactones or volatile phenols are extracted from toasted wood.

The participation of an individual compound to the flavor profile is not only related to its concentration, which can cover wide ranges in distilled spirits, but could also result of its odor perception threshold, defined as the lowest concentration than can be detected by smelling. Therefore, some substances found in liquors present at very low levels contribute significantly to the overall aroma profile of the beverage.

Over the last few years, a large increase in the number of micro-distilleries has been observed. This type of distillery, which is not officially defined, can be described as a small-scale distillery that focuses on developing superior quality products rather than producing high quantities of spirit. Small scale production enables numerous variations at each stage of the process, facilitating the development of innovative fermentation, distillation and aging techniques and offering new aromas and flavors for distilled spirits. Micro-distilleries generally offer diverse types of liquors and several variations of each product [4]. Until recently, however, there has been a lack in chemical analysis of the distilled spirits produced, mostly because it requires efficient, reliable, and fast methods based on expensive apparatus and qualified staff. Consequently, the flavor singularity of their liquors is generally not demonstrated on a chemical point of view.

Currently, the easiest way for small distilleries to analyze their product is to subcontract analyses of distilled spirits to dedicated laboratories. Occasional access to such analytical equipment is possible through agreements and scientific collaboration with academic research laboratories, for example, but some structures might prefer to carry out analysis of their spirits themselves. For those, analytical equipment is now becoming more affordable, a GC equipped with a flame ionization detector (FID) or a single quadrupole GC-MS being, for instance, in the range of 30,000 euros. Obviously, it implies hiring qualified personnel. In every case, proposing a versatile and easy to implement method applicable to different types of spirits, is relevant for private or public laboratories having to develop analytical methods to carry out such analyses and for small distilleries willing to perform analyses of their products on their own.

Chemical analyses of distilled spirits have most frequently been carried out using gas chromatography because it enables the identification of a large number of analytes and can be coupled to various detectors [5]. When a classic chromatographic system is used (GC/FID or single quadrupole GC-MS), direct injection of the spirit enables to detect major components. These compounds may, however, represent a relatively low contribution to the total sensory profile [6,7]. Consequently, a concentration step of the volatile fraction prior to analysis is necessary for the detection of minor compounds. Such volatile fraction concentration is usually achieved using extraction techniques. Because of the chemical diversity and wide concentration range of flavor compounds in distilled spirits, selection and optimization of an appropriate extraction method is critical [8].

Four extraction methods have been mostly employed in the literature for concentrating volatile fraction from distilled spirits, namely liquid/liquid extraction (LLE), solid phase extraction (SPE), solid phase microextraction (SPME) and stir bar sorptive extraction/headspace sorptive extraction (SBSE/HSSE) (see details in the Appendix A). Most of the time, the extraction conditions are optimized using one product for a single spirit type, such as whiskey, rum, gin, etc. Therefore, and because the chemical composition of spirits differs significantly, reported methods have been optimized for one specific distilled spirit only and have not been applied to other types of alcoholic beverages.

Consequently, the aim of the present study was to develop a versatile analytical technique applicable to a large range of distilled liquors and suitable for micro-distilleries. To that end, a model mixture embodying a theoretical distilled spirit was made starting from literature data dealing with the composition of such beverages. A preoptimization study was carried out by submitting the model spirit solution to the four above mentioned extraction techniques, namely LLE, SPE, SPME and SBSE/HSSE, using conditions reported in previous publications. An overall comparison led to the identification of the HS-SPME extraction method as the best compromise according to efficiency and cost criteria. The parameters of this method were then extensively studied, and optimized conditions were applied to three commercial samples of distilled spirits.

## 2. Materials and Methods

### 2.1. Commercial Samples of Distilled Spirits and Chemicals

Rum (Diplomático Selección de Familia) and whiskey (Glenfiddich 14 Year Old Bourbon Barrel Reserve) were bought at a local store. A commercially available gin sample (Comte de Grasse 44° N) was provided by Comte de Grasse. Compounds used to make a model mixture, whose composition will be detailed later, were obtained from Merck (Darmstadt, Germany) with a minimum purity of 96%. Dichloromethane and acetonitrile (Merck) were freshly distilled prior to use. Ethanol (96% purity) was obtained from Isnard (Grasse, France).

### 2.2. Liquid-Liquid Extraction

The model mixture was mixed with deionized water to reduce the ethanol content to 10% (*v*/*v*). Next, 20 g NaCl were dissolved in 200 mL of a sample. Freshly redistilled solvent (3 × 60 mL) was added and left under stirring (500 rpm) for 10 min at room temperature. Solvents tested for extraction of the flavor compounds were as follows: dichloromethane and diethyl ether/pentane 1:1 *v*/*v* mixture. The organic phases were gathered, dried on MgSO_4_, filtrated with a Büchner funnel, and evaporated to 500 µL with a Kuderna–Danish concentrator. The extract was filtrated on a 0.22 µm syringe filter and stored at −18 °C before analysis. Each trial was carried out in triplicate.

### 2.3. Solid Phase Extraction

The model mixture was mixed with deionized water to reduce the ethanol content to 10% (*v*/*v*) and percolated through a SPE cartridge after conditioning with acetonitrile and equilibrating with an ethanol/water 9:1 (*v*/*v*) solution. Cartridges tested for extraction of flavor compounds were as follows: Chromabond C18 (500 mg) and Chromabond hydrophilic-lipophilic balanced HLB (200 mg) (Macherey–Nagel, Düren, Germany), Supelclean LC−8 and LiChrolut EN (Merck, Darmstadt, Germany). Compounds of interest were eluted with dichloromethane (2 × 3 mL), which was evaporated to 500 µL under a nitrogen flow.

### 2.4. Solid Phase Microextraction

Fibers and a manual SPME device were obtained from Supelco Co. (Bellefonte, PA, USA). Fibers tested for extraction of the volatile components were as follows: poly(dimethylsiloxane) (PDMS) 100 μm, carboxen/PDMS (CAR/PDMS) 85 μm and divinylbenzene/CAR/PDMS (DVB/CAR/PDMS) 50/30 μm. Before use, fibers were conditioned as recommended by the manufacturer. To study the influence of ethanol (EtOH) content on the extraction, the model mixture was diluted with deionized water. The EtOH contents (*v*/*v*) studied here were as follows: 40% (undiluted mixture), 10%, 5%. To study the influence of NaCl on the extraction, the following conditions were tested: no salt or 10% NaCl (*w*/*v*), accordingly. Next, 13 mL of sample were then placed in a 40 mL amber vial closed by a polytetrafluoroethylene (PTFE)/silicone septum (Supelco). A water bath was used to control the temperature (20 °C, 30 °C, 45 °C and 60 °C tested) and the sample was conditioned for 5 min before extraction. Each analysis was carried out in triplicate. After exposure, the fiber was thermally desorbed into a GC and left in the injection port (equipped with a 0.75 mm i.d. inlet liner) for 5 min. The injector was set at the temperature recommended by the manufacturer and operated in splitless mode for 5 min.

### 2.5. Stir Bar Sorptive Extraction/Headspace Sorptive Extraction

Extractions were carried out using PDMS commercial stir bars (10 mm length × 0.5 mm film thickness), supplied by Gerstel (Mülheim a/d Ruhr, Germany). A volume of 13 mL of sample diluted to 10%EtOH (*v*/*v*) with deionized water was placed in a 40 mL amber vial. The stir bar was then added to the flask or placed in the headspace. The vial was closed by a PTFE/silicone septum. In SBSE mode, the PDMS stir bar was placed in the liquid and stirred at 800 rpm. In HSSE mode, the PDMS stir bar was placed in the headspace and the sample was stirred at 300 rpm with a standard stir bar. In both cases, the extraction was carried out at 25 °C for 60 min. After removal from the sample, the PDMS stir bar was washed for a few seconds in distilled water and gently dried with a lint-free tissue. It was then transferred into a thermal desorption tube for thermal desorption. Before each extraction, the stir bar was conditioned at 300 °C for 60 min as recommended by the manufacturer.

### 2.6. GC-MS/FID

LLE, SPE and SPME: These analyses were carried out using a 7820A/5977B GC-MS/FID (Agilent, Little Falls, DE, USA) equipped with an HP-5ms capillary column (30 m × 0.25 mm × 0.25 μm, Agilent). The temperature program was as follows: 40 °C for 4 min, 40 to 220 °C at 2 °C/min, 220 to 270 °C at 8 °C/min, holding for 7.25 min. Mass spectroscopy analysis was performed using a source in electron ionization (EI) mode (70 eV ionization energy). Acquisition was performed in scan mode (mass range *m*/*z* 35–350). Identification of the aroma compounds was achieved by comparing the retention indices (RI) and mass fragmented patterns with those of reference compounds or with mass spectrums in commercial (NIST, Wiley) and homemade libraries and previously reported RI in the literature (NIST). The flame ionization detector (FID) was used at 270 °C with H_2_ at 40 mL/min and air at 400 mL/min. Linear retention indices of the compounds were calculated using an n-alkane series.

SBSE/HSSE: Coated stir bars were thermally desorbed using an ATD-350 thermodesorption system (Perkin Elmer, Waltham, MA, USA) at 250 °C for 15 min under a helium flow (50 mL min^−1^) and desorbed analytes were focused on a trap at −35 °C. Finally, the trap was programmed from −35 °C to 300 °C (held for 3 min) at 20 °C s^−1^ for analysis by GC-MS. Capillary GC-MS analyses mode were performed using an Clarus 680-SQ 8T system (Perkin Elmer, Waltham, MA, USA), equipped with an Elite-5ms capillary column (30 m × 0.25 mm × 0.25 μm, Perkin Elmer, Waltham, MA, USA). The temperature program was as follows: 40 to 220 °C at 2 °C/min, 220 to 270 °C at 8 °C/min, holding for 6.25 min. Mass spectrum analysis was performed using a source in electron ionization (EI) mode (70 eV ionization energy). Acquisition was performed in scan mode (mass range *m*/*z* 35–350). Identification of aroma compounds of the model mixture was achieved by comparing mass fragmented patterns with mass spectrums in the NIST commercial library and confirmed by injection of chemical standards.

### 2.7. Statistical Analysis

All analyses were carried out in triplicate. Statistical analysis was performed using Microsoft Excel version 365 (Microsoft Corporation, Redmond, WA, USA). Concerning SPME temperature and time optimization, mean values and standard deviations were calculated. Statistical significance was determined by one-way analysis of variance (ANOVA). Chemical analysis for each commercial sample was also carried out injecting three times each sample in the GC-MS apparatus. All three chromatograms were analyzed separately, and the abundance of each compound identified on the chromatogram was given according to the following abundance scale: “+++++”: 60% > *x* ≥ 40%; “++++”: 40% > *x* ≥ 20%; “+++”: 20% > *x* ≥ 1%; “++”: 1% > *x* ≥ 0.1%; “+”: 0.1% > *x* ≥ 0.01%; “tr”: *x* < 0.01%. For the same distilled spirit, each compound was found on all three chromatograms with the same abundance range.

## 3. Results and Discussion

### 3.1. Model Spirit Solution

In order to make a model mixture representative of a large range of spirit drinks, 13 articles relating quantitative analyses of 87 commercial samples of various distilled spirits, including whiskey, bourbon, gin, vodka, brandy, rum, cachaça, mezcal, pear spirit, and baijiu, were studied [7,9,10,11,12,13,14,15,16,17,18,19,20]. It led to the identification of almost 300 compounds usually present in liquors and representative of the flavor diversity in distilled spirits. These molecules were classified in 15 chemical classes and, for each class, at least one of the most recurrent compounds was selected. Additional compounds were added to study a possible influence of the chemical structure on the extraction such as chain length (ethyl hexanoate/ethyl decanoate), aromaticity (decanol/2-phenylethanol and decanal/benzaldehyde) and presence of a cycle (2-octanone/(E)-β-damascenone). Twenty-four compounds (including the (E) and (Z) isomers of whiskey lactone), listed in Table 1, were finally chosen. Each selected compound was put in the ethanol/water (40:60 *v*/*v*) model mixture at its mean concentration observed in all distilled spirits, a value calculated thanks to the literature study. When several liquors of the same type were analyzed in one article, only the lowest and highest concentration values for each compound were taken into account, without considering the cases in which a compound was not detected. As a result, each calculated concentration was the average of one to twenty-seven values. Within the model mixture, sotolon and furfural exhibited very different concentration values, namely 0.00093 and 18.2 mg·L^−1^, respectively, corresponding to a concentration ratio of over 20,000. In addition, it was also interesting to note that six out of twenty-four compounds, namely diethyl succinate, sotolon, 1,1,3-triethoxypropane, 2,3,5-trimethylpyrazine, o-xylene and (Z)-nerolidol, had mean concentrations below their mean odor thresholds in water. Nevertheless, literature reported that synergetic effects occur with compounds at concentrations below their odor thresholds leading to their participation to the flavor profile [21].

### 3.2. Extraction Technique Pre-Optimization and Comparison

The model mixture was then submitted to all four selected extraction techniques detailed above and analyzed using GC-MS/FID. For each method, the most common conditions reported in previous publications for the extraction of distilled spirits were tested (see details in the Appendix A). The main parameters were pre-optimized by studying the number of detected compounds and their total area. The amount of model mixture, injection volume and split ratio were adapted for each method to have comparable conditions.

The two most widely used solvents reported for liquid/liquid extraction of distilled spirits in the literature are dichloromethane and diethyl ether/pentane mixtures. The best results obtained using dichloromethane and diethyl ether/pentane were 23 and 22 detected compounds out of 24, respectively. Under our conditions, hexanal could not be detected using diethyl ether/pentane, probably because of its low concentration in the model mixture. With both solvents, a salting-out effect could be observed because the addition of NaCl enabled to detect dimethyl trisulfide.

Solid phase extraction of the model mixture was first carried out with two reversed-phase silica sorbents (C18, C8). It led to the detection of 12 and 13 compounds, respectively. When polymeric sorbents were used (LiChrolut EN, Strata-X, HLB), 16, 21 and 22 compounds, respectively, were detected. These results can be explained by their higher inner surface and capacity to retain a wide range of analytes [24]. Ethylvinylbenzene-divinylbenzene copolymer (LiChrolut EN) allowed for the detection of 16 compounds. This is surprising considering the fact that it is the most commonly used sorbent for the extraction of aroma compounds of various liquors [25,26,27]. However, in these studies, LiChrolut EN was reported to exhibit a selectivity for certain compounds. The influence of the pH on the retention and selectivity of the LiChrolut EN sorbent for the extraction of aroma compounds can also explain a compound detection relatively lower than expected [28]. While a pH of 7.9 was measured for the model mixture, distilled spirits are generally acidic with measured pH values of 7.1, 4.3 and 4.1 for commercial samples of gin, whiskey, and rum, respectively, studied here. An experiment carried out with a pH 4 buffer solution led to the detection of two additional compounds, namely decanoic acid and decanol. Acidification of the solution might, however, have induced chemical reactions between the components of the mixture, decanoic acid being possibly obtained by hydrolysis of ethyl decanoate. For that reason, the model mixture was not acidified in the rest of the study. To the best of our knowledge, extraction of distilled spirits using Strata-X has never been carried out so far, whereas HLB sorbent use has only been reported once [29]. Because N-vinylpyrrolidone-divinylbenzene copolymer (HLB) allowed for the detection of 22 compounds out of 24, it was identified as the most suitable polymeric sorbent for solid-phase extraction of liquors.

Preliminary solid phase microextraction experiments were carried out to evaluate the influence of fiber type (PDMS, CAR/PDMS, DVB/CAR/PDMS), extraction mode with a PDMS fiber (headspace and immersion), ethanol strength (5, 10, and 40% EtOH *v*/*v*) and addition of NaCl. For this study, extraction temperature and time were set to 20 °C and 60 min, respectively. The PDMS liquid polymer coating, whose non-competitive extraction mechanism enables quantitative analyses, was tested in the first place. Liquid immersion and headspace extraction only led to the detection of 17 and 18 compounds, respectively, so mixed phase fibers were tested. These coatings are known to offer higher sensitivity but competition between the analytes can be observed because of their extraction through adsorption. In headspace mode, CAR/PDMS and DVB/CAR/PDMS fibers led to the detection of 20 and 21 compounds, respectively. This is in accordance with literature as DVB/CAR/PDMS is the preferred fiber type for the analysis of distilled spirits [18,19,30,31]. With the DVB/CAR/PDMS fiber, reducing the ethanol strength from 40 to 10% *v*/*v* increased the number of compounds detected by limiting the adsorption of ethanol onto the fiber [32]. However, a higher dilution (5% EtOH *v*/*v*) led to a lower number of detected compounds. This was most probably because the concentrations of aroma compounds in the sample were too low. Addition of sodium chloride led to an increase in the number of detected compounds, revealing a non-negligible salting-out effect.

The stir bar and headspace sorptive extractions of the model mixture at room temperature during 60 min gave very similar results in terms of detected compounds (22 in both cases) and total TIC areas. As a consequence, their efficiencies could not be compared using the model mixture but several previous reports showed that SBSE showed higher sensitivity than HSSE for the analysis of alcoholic beverages [33,34]. Increasing the extraction time to 6 h did not enable to detect additional compounds but it doubled and tripled the total TIC area for SBSE and HSSE, respectively.

Among all four tested extraction techniques, LLE gave the best results (Table 2). Only one compound was not detected, likely because of its too low concentration (sotolon—0.000938 mg/L). However, this method presents some disadvantages. It is time-consuming because a slow evaporation rate is used with the Kuderna–Danish concentrator to prevent the loss of highly volatile compounds. This could, however, be avoided using nitrogen flow or SpeedVac concentrators. Because 50 and 200 mL of sample and solvent, respectively, were needed, this technique appears to be consuming in terms of sample and solvent, leading to potential environmental and health issues. Finally, a poor repeatability can be observed with this technique.

The efficiency of SPE in terms of compounds detected was slightly lower than that of LLE, only (Z)-nerolidol being not detected. In our case and despite a lower solvent consumption, this technique offered relatively low benefits compared to LLE because the extractions had to be carried out manually and the same amount of sample was necessary. Single use only cartridges present an important advantage from an analytical point of view but have a significant impact on the environment.

Although it was not in this study, SPE can be automated either in off-line or on-line mode [35]. The off-line approach offers operational flexibility and equipment simplicity, but still requires an important handling time and is not applicable to high sampling frequencies. On-line SPE, in which analytes are directly transferred to the analytical column, offers a low time consumption and a relatively low cost per analysis. However, skilled personnel are required and manual SPE methods are not easily transferred to on-line mode. In particular, the amount of solid phase in precolumns used for on-line SPE is generally lower than in off-line cartridges. In addition, the analytes being eluted and injected into the GC use the same solvent, so the latter must be adapted to the chromatographic conditions. Finally, the cost of this kind of equipment is high and could be incompatible with the budget of micro-distilleries.

The SPME was slightly less efficient than the LLE and SPE extraction methods (21 compounds detected). 1,1,3-triethoxypropane could not be detected with any of the three tested fiber types, possibly because of its low affinity for relatively apolar fibers. However, SPME offers considerable advantages to micro-distilleries. In manual mode, this method is easy to operate and only requires basic and inexpensive consumables. Full automation is possible at a relatively affordable price compared to on-line SPE and operating parameters can be effectively transposed from manual to automated mode without major adjustment. In that case, a very low handling time is necessary. Moreover, SPME fibers can be thermally desorbed directly in the GC injector and do not require specific equipment.

The SBSE and HSSE extraction methods gave slightly better results than SPME with 22 detected in both immersion and headspace modes. This can be explained by the fact that stir bars have a much larger coating volume than SPME fibers. Neither with SBSE nor HSSE was decanoic acid detected despite its relatively high concentration. This might be a consequence of its low affinity with the apolar poly(dimethylsiloxane) stir bar in addition to the poor response of fatty acids with apolar columns. In contrast to SPME, stir bar extractions have to be carried out manually [36] and require a specific equipment for thermal desorption and, therefore, a supplementary cost.

Even though LLE appears to be the most efficient technique for the extraction of distilled spirits in terms of detected compounds, SPE, SPME and SBSE/HSSE present some advantages. To identify the most efficient technique, two situations can be envisaged.

If a quantitative analysis is desired, LLE and SPE are the most appropriate methods. The results of this study showed that LLE enabled to detect more compounds of the model mixture. The main advantage of SPE is its possible full automation, but the cost of this equipment and the need for skilled operators might be a limitation of this technique for micro-distilleries.

Even if quantitative analyses bring valuable information about the flavor profile of distilled spirits, they are not always necessary. In particular, a large quantity of trials can be done by micro-distilleries while developing new products. In that situation, qualitative analyses bringing quick results at an affordable price in of high interest. This study showed that SPME offers the best compromise in terms of efficiency, cost, solvent and sample consumption, and environmental impact.

Consequently, SPME was selected for further optimization. Using this method, the best results were obtain using a DVB/CAR/PDMS fiber and addition of NaCl. The influences of extraction time and temperature on peak area were then studied.

### 3.3. SPME Time and Temperature Optimization

To identify the optimal conditions in terms of time and temperature for the extraction of the model mixture, the TIC peak areas of four compounds representative of the chemical diversity of the model mixture, namely α-pinene, furfural, β-caryophyllene and (E)-β-damascenone were specifically studied. Because extraction of volatile compounds onto an SPME fiber is influenced by their volatility and their polarity, these four compounds were selected according to these properties (Table 3). On the one hand, α-pinene and furfural were chosen because they have two of the lowest boiling points among all molecules present in the model mixture and because they are relatively apolar and polar, respectively. On the other hand, β-caryophyllene and (E)-β-damascenone both exhibit high boiling points but have different polarities. Despite its higher polarity and boiling point than (E)-β-damascenone, vanillin’s TIC peak area is not represented here because of a partial coelution, but its SIM area (*m*/*z* 152) exhibited the same behavior as (E)-β-damascenone’s one.

The extraction temperature was optimized by following the TIC signal of the four representative compounds at 20 °C, 30 °C, 45 °C, and 60 °C to maximize their peak areas (Figure 1). Increase in temperature is known to have two antagonistic effects on the extraction: it increases analytes’ concentration in the headspace as well as their diffusion coefficient, but also decreases the distribution constant between the sample and the fiber [38].

For furfural, α-pinene and β-caryophyllene as well as the total area of the chromatogram, the highest peak area was obtained at an extraction temperature of 30 °C. Because of its low volatility, a low concentration in the model mixture (0.0469 mg/L), and a presumably low affinity for the SPME coating, (E)-β-damascenone was particularly difficult to detect as its peak area was much lower than those of other compounds. Because the largest peak area for (E)-β-damascenone was found to be slightly higher at 45 °C than at 30 °C, an extraction temperature of 35 °C was chosen as the best compromise for the detection of the four compounds.

Extraction kinetics were studied by measuring the peak areas after 5, 30, 60, and 120 min of extraction. As shown in Figure 2, at least 60 min of extraction were necessary to reach equilibrium for α-pinene and furfural. With a decreasing peak area over time, α-pinene presented a singular kinetic profile. This decrease in peak area over time can be explained by a competition occurring between compounds for their adsorption onto the fiber. Because of its high volatility, the concentration of α-pinene in the headspace is expected to be high. α-pinene is therefore extracted rapidly by the sorbent. Nevertheless, over time, α-pinene might be replaced onto the fiber by other less volatile compounds exhibiting a higher affinity for the fiber, leading to a desorption of α-pinene and a decrease in α-pinene’s signal during extraction. In the case of β-caryophyllene, equilibrium was not reached during extraction, but peak areas obtained at 60 and 120 min were close. With an extraction time of 60 min, this molecule showed a signal more intense than all three other compounds and was easily detected. On the contrary, the peak area of (E)-β-damascenone was much lower during the entire extraction. After 60 min of extraction, the signal for (E)-β-damascenone was acceptable. To obtain the best compromise between a satisfactory signal for all four studied compounds and a cost-effective extraction method, an extraction time of 60 min was chosen.

Overall, the optimum conditions for the extraction of volatile compounds from a distilled spirit using HS-SPME were found to be as follows. First, the sample was diluted to reach a concentration of ethanol of 10% (*v*/*v*) and 10% NaCl (*w*/*v*) added. Extraction was then carried out using a DVB/CAR/PDMS fiber, at 35 °C, with an extraction time of 60 min.

### 3.4. Analysis of Gin, Rum, and Whiskey Samples by HS-SPME-GC-MS

To investigate its versatility and its ability to be applied to distilled spirits of significantly different nature, the HS-SPME method was applied to the analysis of real samples. To that end, commercially available gin, rum, and whiskey, respectively 44° N from Comte de Grasse, Seleccion de Familia from Diplomático, and 14 Year Old Bourbon Barrel Reserve from Glenfiddich, were selected. These three distilled spirits exhibit distinct sensory profiles and production processes.

The extraction of headspace was carried out using the above-mentioned method and followed by a GC-MS analysis as detailed in the experimental section. Results are collected in Table 4. Chromatograms for each spirit revealed that 130, 75 and 59 compounds were identified for gin, rum, and whiskey, respectively. Combining results from the three spirits used here, a total of 215 compounds were detected. Among them, 188 were identified as alcohols, esters, lactones, carbonyls, acetals, fatty acids, phenols, furans, aromatics, monoterpenes, oxygenated monoterpenes, sesquiterpernes, oxygenated sesquiterpenes, diterpenes, oxygenated diterpenes, alkenes, and alkanes. As usually reported, compounds described as unknown were not identified because their mass fragmentation patterns (see details in the Appendix A) were not registered in all databases available during this study. The signal intensity recorded on chromatograms followed the same order as the number of compounds detected: gin’s total area (TIC) was twice as high as that of the whiskey’s one and 20 times as high as that of rum. Totals of 121, 33 and 18 compounds were found to be specific to gin, whiskey, and rum, respectively. While 39 compounds were found both in rum and whiskey, only four molecules, namely ethyl hexanoate, ethyl hexadecanoate, nonanal, and limonene, were common to the three spirits. This was expected because of their production methods: gin is produced through maceration and/or distillation of botanicals in a water-ethanol mixture, while rum and whiskey are obtained by fermenting and distilling saccharide-rich raw materials and underwent oak aging in this case.

In the case of gin, terpenoids represent the main chemical class found with 99 different compounds including 18 monoterpenes, 27 oxygenated monoterpenes, 31 sesquiterpenes, 20 oxygenated sesquiterpenes, 2 diterpenes, and 1 oxygenated diterpene. As reported previously, the mass spectra of these kinds of molecules being very similar, 16 of them could not be identified [39]. Terpenoids originate from thirteen botanicals and eight botanical extracts used for the production of this commercial gin, including juniper berries which is the major source of flavors in gin according to the EU regulation. Among terpenoids found in this sample, α-pinene (piney, woody), β-pinene (woody, cooling), β-myrcene (woody, rose, peach), limonene (lemon, orange, sweet), β-caryophyllene (spicy, woody) and α-humulene (balsamic, flowery, grassy) are key flavor constituents of juniper berries [40] and represent high relative peak areas of the chromatogram. Some of them are also abundant in other botanicals such as limonene which is a main constituent of citrus essential oils. Esters are the second most represented chemical class in gin with nine compounds identified.

In the case of whiskey and rum, ester is the chemical class most represented in these spirits with 43 and 31 compounds, respectively. Esters contribute positively to the flavor profile by conferring fruity and floral notes to the liquor. This result is in agreement with previous reports [20,41,42]. Alcohols represent the second largest group in terms of number of components identified in whiskey and rum with nine and five compounds, respectively, all alcohols identified in rum being also present in the whiskey sample. Most of these compounds have characteristic fusel and alcoholic aromas but can contribute positively to the overall flavor profile at low concentrations [42]. In addition to esters and alcohols, several important flavor compounds were identified in both whiskey and rum. (E)-whiskey lactone, which originates from aging in oak casks, provides woody and coconut aromas to the liquor. (Z)-whiskey lactone is probably present in both samples but could not be definitely identified because of coelution. Carbonyls like benzaldehyde and β-damascenone are responsible for almond and woody notes, respectively. Furfural, a furan that can be formed through Maillard reaction and caramelization [43], has a sweet and caramellic aroma.

Despite similar compositions, each of the two spirits’ specificities could be observed. On the one hand, the presence of 2-phenylethanol, which has a typical rose flavor, was only observed in whiskey. Interestingly, its corresponding ester 2-phenylethyl acetate was detected both in whiskey and rum. Two fatty acids, namely octanoic and dodecanoic acid, were also specific to whiskey, as well as hexyl acetate whose fruity flavor is described as apple and pear. On the other hand, minty flavor compounds were identified in rum with the presence of menthol, methyl salicylate and ethyl salicylate. Moreover, it appeared that rum was the only sample in which phenols were detected. In particular, dihydroeugenol is responsible for clove and peppery notes. The presence of allyl isothiocyanate in the rum sample was not expected, this compound being a constituent of Brassicaceae species. It is also known to be used after sugarcane harvesting to prevent losses in weight and sugar due to natural dissipation [44].

Finally, even though whiskey and rum have been oak-aged for several years, the number of phenols and furans identified in these two spirits was somehow lower than expected. The low affinity of the DVB/CAR/PDMS fiber for relatively polar compounds could explain this phenomenon. This might be a limitation of this method because these compounds are known to provide warm notes to oak-aged liquors. Not detecting such compounds might result in a loss of information on the flavor profile.

Overall, it appears that the HS-SPME method optimized here was particularly efficient at detecting many molecules from three different types of distilled spirits. This technique enabled to identify 188 volatile compounds representing 18 chemical families. Notably, 99 terpenoids and 53 esters were identified. In addition, the detection of distinct compounds allowed to discriminate between the three liquors analyzed here. Therefore, the use of a model mixture for the development of the extraction method appeared to be an efficient strategy. By comparison, a previous report describing the optimization of HS-SPME conditions with a single brandy sample led to the identification of 158 compounds when it was applied to six different distilled spirits, namely brandy, whiskey, rum, Chinese liquor, and gin [20].

Furthermore, the HS-SPME method also allowed for the identification of compounds never identified before in a spirit. As an example, to the best of our knowledge, it is the first time (Z)-3-hexenyl benzoate (green, floral) has been identified in gin.

Compared to other methods discussed in the manuscript and in the literature, HS-SPME appears to be a very versatile, easy to implement extraction method for a fast and reliable qualitative analysis of distilled spirits. According to our study, the HS-SPME-GC-MS technique is fast, low solvent and sample consuming, relatively affordable, and effective for the detection of a wide range of flavor compounds in distilled spirits. This method is particularly adapted to micro-distilleries, that can, for a limited investment in apparatus and technical skills, provide a reliable way of analyzing their samples and comparing them to other products.

## 4. Conclusions

In this study, a model mixture embodying a theoretical distilled spirit was prepared. It highlighted the diversity and the wide range of concentrations of flavor compounds found in liquors.

When applied to the model liquor, the HS-SPME extraction method was preferred to the LLE, SPE and SBSE/HSSE methods. A study showed that the extraction was optimal with a DVB/CAR/PDMS fiber, during 60 min, at 35 °C and with the addition of NaCl. As shown in previous reports, a decrease of the ethanol content to 10% *v*/*v* was necessary to improve the method’s efficiency.

The optimized HS-SPME method was then successfully applied to gin, whiskey and rum commercial samples and led to the identification of 188 flavor compounds. While terpenoids were the most abundant compounds in the gin sample, esters were the most represented chemical family in whiskey and rum. The method enabled to differentiate the compositions of the three spirits, as 121, 33 and 18 compounds were found to be specific to gin, whiskey, and rum, respectively.

Finally, all four extraction techniques studied in this work showed limitations. Notably, even though the HS-SPME method developed here with a DVB/CAR/PDMS fiber provides substantial information about the flavor compounds in distilled spirits at a price appropriate for micro-distilleries, it has some disadvantages. It is not quantitative, and the extraction time is not negligible. Moreover, it appeared to be less efficient for the extraction of more polar compounds such as phenols and furans, possibly resulting in a loss of information on the flavor profile. However, when a standard chromatographic system like a single quadrupole GC-MS or GC/FID is used, this extraction step is necessary.

Overall, the HS-SPME method developed here appears to be well-suited for the analysis of several distilled spirits and could be used by micro-distilleries eager to control their production, either by carrying out themselves the analysis on an affordable GC-MS equipment or by subcontracting analysis to a laboratory that will be able to use the method.

Evaluation of the HS-SPME extraction method to identify compounds present in other types of distilled spirits such as vodka, tequila or brandy will be undertaken soon. In addition, this method will be used to analyze different brands of the same distilled spirit to highlight chemical specificities of a given product. In parallel, the results reported here using the HS-SPME method and direct injection without any preliminary concentration into a more sensitive chromatographic system, such as GCxGC-TOF-MS, will be compared.

## Figures and Tables

**Figure 1 foods-11-03358-f001:**
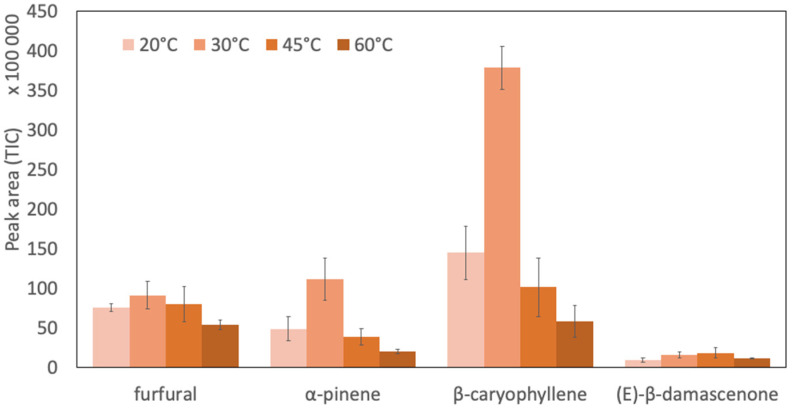
Impact of temperature on the peak area (TIC) with standard deviations of four selected compounds of the model mixture. (*n* = 3).

**Figure 2 foods-11-03358-f002:**
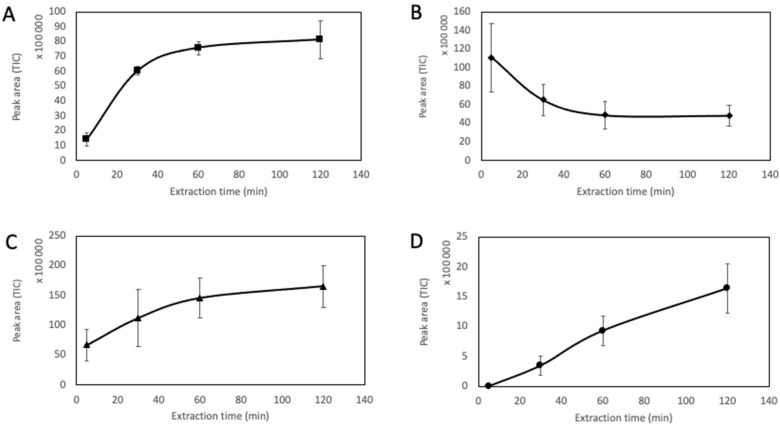
Extraction time curves with standard deviations of four selected compounds of the model mixture (*n* = 3). (**A**) furfural; (**B**) α-pinene; (**C**) β-caryophyllene; (**D**) (E)-β-damascenone.

**Table 1 foods-11-03358-t001:** Composition of the model mixture.

Chemical Class	Compound	CAS	Concentration (mg/L)	Mean Odor Threshold in Water (mg/L) ^a^
Alcohol	2-phenylethanol	60-12-8	7.31	1.331
	decanol	112-30-1	0.114	0.047
Ester	ethyl hexanoate	123-66-0	17.4	0.00186
	diethyl succinate	123-25-1	3.78	353 ^b^
	2-phenylethyl acetate	103-45-7	0.457	0.249
	ethyl decanoate	110-38-3	7.45	1.299
Lactone	sotolon	28664-35-9	0.000938	0.00565
	(Z)-whiskey lactone	80041-00-5	0.69	0.035 ^c^
	(E)-whiskey lactone	39638-67-0	0.31	0.05 ^d^
Carbonyl	hexanal	66-25-1	0.132	0.0501
	benzaldehyde	100-52-7	7.98	1.72
	2-octanone	111-13-7	0.261	0.0418
	(E)-β-damascenone	23726-93-4	0.0469	0.00125
Acetal	1,1,3-triethoxipropane	7789-92-6	0.103	3.70 ^b^
Acid	decanoic acid	334-48-5	5.56	5.464
Phenol	vanillin	121-33-5	2.86	0.301
Furan	furfural	98-01-1	18.2	9.602
Pyrazine	2,3,5-trimethylpyrazine	14667-55-1	0.154	0.216
Sulfur compound	dimethyl trisulfide	3658-80-8	0.0595	0.00003
Aromatic	o-xylene	95-47-6	0.313	0.45
Monoterpene	α-pinene	80-56-8	4.02	0.296
Oxygenated monoterpene	linalool	78-70-6	5.28	0.0108
Sesquiterpene	β-caryophyllene	87-44-5	0.679	0.461
Oxygenated sesquiterpene	(Z)-nerolidol	3790-78-1	0.0387	0.1
Total	24 compounds			

^a^ from [22] ^b^ in a 46% (*v*/*v*) ethanol solution, from [23] ^c^ in a 12% (*v*/*v*) ethanol solution ^d^ in a 9.4% (*w*/*w*) grain spirit.

**Table 2 foods-11-03358-t002:** Results obtained by applying different extraction techniques to the model mixture.

Extraction Technique	LLE	SPE	SPME	SBSE	HSSE
Detected compounds (/24)	23	22	21	22	22
Handling time *	3 h	1 h	0.2 h	0.2 h	
Sample amount	50 mL	13 mL	3 mL	3 mL	
Solvent amount	200 mL	10 mL	-	-	
Quantitative analysis	+++	+++	+	+	
Automation	Low	High	High	Moderate	
Cost per analysis	Low	Low	Moderate	Moderate	
Cost of instrumentation	Low	High	Moderate	High	

* in this study, all extractions were carried out manually. “+++”: 20% > *x* ≥ 1%; “+”: 0.1% > *x* ≥ 0.01%. “-“: not relevant.

**Table 3 foods-11-03358-t003:** Selected properties of the four studied compounds.

	Boiling Point (°C)	Log P (o/w)
α-pinene	155–156	4.830 ^a^
furfural	161–162	0.410 ^a^
β-caryophyllene	254–257	6.777 ^a^
(E)-β-damascenone	274–276	4.120 ^b^

^a^ from The Good Scents Company Information System. ^b^ from [37].

**Table 4 foods-11-03358-t004:** Volatile compounds by chemical class identified in 3 commercial liquors analyzed by HS-SPME-GC-MS.

Chemical Class	Compound	exp. RI	th. RI ^a^	Gin ^b^	Rum ^b^	Whiskey ^b^
alcohol	2-methyl-1-propanol	623	624	-	+	+
	3-methyl-1-butanol	736	736	-	++	+
	2-methyl-1-butanol	738	739	-	++	+
	octanol	1068	1070	-	tr	tr
	2-phenylethanol	1109	1116	-	-	+
	decanol	1268	1272	-	-	+
	dodecanol	1452	1474	-	tr	+
	tetradecanol	1678	1676	-	-	+
	hexadecanol	1870	1880	-	-	tr
ester	ethyl acetate	611	612	-	+	+
	ethyl propanoate	716	710	-	tr	tr
	ethyl 2-methylpropanoate	754	756	-	-	tr
	2-methylpropyl acetate	771	772	-	tr	tr
	ethyl butanoate	801	802	-	tr	tr
	ethyl 2-methylbutanoate	847	849	-	-	tr
	ethyl 3-methylbutanoate	849	853	-	tr	tr
	3-methylbutyl acetate	873	872	-	+	+
	2-methylbutyl acetate	875	879	-	tr	tr
	ethyl (E)-2-methyl-2-butenoate	937	939	-	-	tr
	ethyl hexanoate	995	999	tr	++	+
	hexyl acetate	1010	1011	-	-	tr
	ethyl heptanoate	1094	1098	-	+	tr
	3-methylbutyl angelate	1144	1153	tr	-	-
	2-methylpropyl hexanoate	1146	1149	-	-	tr
	benzyl acetate	1153	1164	+	-	-
	ethyl benzoate	1157	1172	-	tr	tr
	diethyl succinate	1169	1181	-	tr	-
	ethyl octanoate	1195	1196	-	++++	+++
	octyl acetate	1206	1210	tr	-	tr
	3-methylbutyl hexanoate	1238	1250	-	tr	tr
	ethyl phenylacetate	1239	1247	-	-	tr
	2-phenylethyl acetate	1244	1258	-	tr	+
	2-methylbutyl hexanoate	1249	1247	-	-	tr
	ethyl nonanoate	1284	1295	-	+	+
	2-methylpropyl octanoate	1344	1348	-	-	tr
	ethyl 9-decenoate	1370	1388	-	++	+
	ethyl decanoate	1392	1396	-	+++++	+++++
	decyl acetate	1421	1409	-	-	+
	3-methylbutyl octanoate	1424	1446	-	+	+
	2-methylbutyl octanoate	1427	1449	-	+	+
	ethyl undecanoate	1473	1494	-	tr	+
	propyl decanoate	1491	1492	-	-	tr
	2-methylpropyl decanoate	1523	1546	-	+	+
	methyl dodecanoate	1524	1526	-	-	tr
	(Z)-3-hexenyl benzoate	1568	1570	tr	-	-
	ethyl dodecanoate	1599	1594	-	++	++++
	dodecyl acetate	1614	1607	-	-	tr
	2-phenylethyl hexanoate	1640	1650	-	tr	tr
	3-methylbutyl decanoate	1645	1645	-	++	+
	2-methylbutyl decanoate	1651	1647	-	tr	+
	propyl dodecanoate	1692	1685	-	-	tr
	ethyl tridecanoate	1696	1687	-	-	tr
	methyl tetradecanoate	1722	1725	tr	-	-
	benzyl benzoate	1757	1763	tr	-	-
	octyl octanoate	1776	1779	-	tr	-
	ethyl tetradecanoate	1794	1793	-	+	+
	isopropyl myristate	1824	1825	tr	+	-
	2-phenylethyl ester + 3-methylbutyl dodecanoate	1842	- + 1847	-	+	-
	2-phenylethyl octanoate	1846	1851	-	-	tr
	methyl hexadecanoate	1924	1926	tr	-	-
	ethyl 9-hexadecenoate	1975	1976	-	-	+
	ethyl hexadecanoate	1993	1993	tr	+	tr
lactone	γ-butyrolactone	913	916	-	-	tr
	(E)-whiskeylactone	1308	1302	-	tr	tr
carbonyl	hexanal	799	801	-	-	tr
	benzaldehyde	953	962	-	tr	tr
	2-nonanone	1086	1092	-	-	tr
	nonanal	1098	1104	tr	+	tr
	decanal	1192	1206	tr	+	-
	2-decanone	1285	1294	+	-	-
	2-undecanone	1288	1294	-	-	+
	undecanal	1301	1307	-	-	tr
	β-damascenone	1365	1386	-	+	tr
acetal	diethoxymethane	700	658	-	tr	-
	1,1-diethoxyethane	724	728	tr	-	tr
	1,1-diethoxybutane	921	901	-	tr	tr
acid	acetic acid	609	610	tr	-	-
	octanoic acid	1206	1180	-	-	+
	dodecanoic acid	1594	1567	-	-	+
phenol	methyl salicylate	1176	1192	-	+	-
	ethyl salicylate	1248	1270	-	tr	-
	dihydroeugenol	1337	1373	-	tr	-
	dillapiol	1621	1622	tr	-	-
furan	furfural	827	833	-	+	+
	4,7-dimethylbenzofuran	1200	1220	tr	-	-
	vitispirane	1272	1271	-	-	+
aromatic	toluene	758	763	tr	-	tr
	ethylbenzene	856	855	-	-	tr
	styrene	885	893	-	tr	tr
	1,2,4-trimethylbenzene	987	990	-	-	tr
	cadalene	1671	1674	+	-	-
	1,4-dimethyl-7-(1-methylethyl)-azulene	1769	1775	tr	-	-
	(1-methyldodecyl)benzene	1909	1916	-	-	tr
monoterpene	α-thujene	920	929	+	-	-
	α-pinene	929	937	++	-	-
	dehydrosabinene	937	956	tr	-	-
	camphene	939	952	tr	-	-
	thuja-2,4(10)-diene	945	956	+	-	-
	sabinene	965	974	+	-	-
	β-pinene	967	979	+	-	-
	β-myrcene	989	991	++	-	-
	α-phellandrene	997	1005	+	-	-
	α-terpinene	1009	1017	+	-	-
	p-cymene	1018	1025	+	-	-
	limonene	1023	1030	++	tr	tr
	(Z)-β-ocimene	1031	1038	+	-	-
	(E)-β-ocimene	1041	1049	+	-	-
	ɣ-terpinene	1052	1060	++	-	-
	terpinolene	1080	1088	+	-	-
	perillene	1092	1101	+	-	-
	p-mentha-1,3,8-triene	1101	1119	tr	-	-
oxygenated monoterpene	linalool	1094	1099	+	-	-
	1-octen-3-yl acetate	1105	1111	+	-	-
	ɑ-pinene epoxide	1116	1095	+	-	-
	unknown 3	1121	-	+	-	-
	pinocarveol	1127	1138	tr	-	-
	camphor	1132	1144	tr	-	-
	citronellal	1143	1153	tr	-	-
	unknown 4	1147	-	+	-	-
	unknown 5	1151	-	+	-	-
	menthol	1159	1170	-	+	-
	isopinocamphone	1161	1173	tr	-	-
	terpinen-4-ol	1167	1177	+	-	-
	verbenyl ethyl ether	1174	1186	+	-	-
	α-terpineol	1180	1189	tr	-	-
	piperitol	1206	1208	+	-	-
	fenchyl acetate	1209	1224	tr	-	-
	citronellol	1221	1228	+	-	-
	thymol methyl ether	1226	1235	++	-	-
	carvacrol methyl ether	1233	1244	tr	-	-
	carvotanacetone	1236	1246	tr	-	-
	linalyl acetate	1247	1257	+	-	-
	geranial	1261	1270	tr	-	-
	bornyl acetate	1277	1285	++	-	-
	lavandulyl acetate	1283	1289	+	-	-
	myrtenyl acetate	1315	1327	+	-	-
	citronellyl acetate	1346	1353	+	-	-
	neryl acetate	1360	1365	++	-	-
	geranyl acetate	1381	1282	++	-	-
sesquiterpene	unknown 8	1321	-	+	-	-
	unknown 10	1329	-	+	-	-
	α-cubebene	1342	1351	++	-	-
	α-ylangene	1363	1372	+	-	-
	α-copaene	1369	1376	++	-	-
	β-elemene	1389	1391	++	-	-
	isolongifolene	1396	1391	+	-	-
	β-caryophyllene	1419	1425	++	-	-
	β-cubebene	1424	1433	+	-	-
	ɣ-elemene	1434	1440	++	-	-
	β-farnesene ^c^	1435	1457	++	+	-
	alloaromadendrene	1436	1448	+	-	-
	α-humulene	1456	1457	++	-	-
	β-farnesene	1459	1457	++	-	-
	germacrene D	1461	1472	+	-	-
	α-elemene	1471	1469	+	-	-
	ɣ-muurolene	1481	1483	++	-	-
	α-curcumene	1484	1486	++	-	-
	β-selinene	1488	1486	++	-	-
	valencene	1492	1492	+	-	-
	curzerene	1497	1498	++	-	-
	α-muurolene	1503	1499	++	-	-
	β-cadinene	1507	1518	+	-	-
	ɣ-cadinene	1519	1513	++	-	-
	δ-cadinene	1531	1524	++	-	-
	cadina-1,4-diene	1537	1532	++	-	-
	selina-3,7(11)-diene	1540	1542	+	-	-
	α-cadinene	1543	1538	++	-	-
	α-calacorene	1547	1542	++	-	-
	germacrene B	1558	1557	+	-	-
	dihydroneoclovene	1633	1680	+	-	-
oxygenated sesquiterpene	unknown 11	1530	-	-	+	-
	unknown 14	1562	-	+	-	-
	1,5-epoxysalvial-4(14)-ene	1565	1573	+	-	-
	spathulenol	1575	1577	tr	-	-
	unknown 16	1578	-	+	-	-
	caryophyllene oxide	1581	1581	+	-	-
	unknown 17	1582	-	+	-	-
	unknown 18	1585	-	+	-	-
	unknown 19	1587	-	+	-	-
	salvialenone	1591	1595	+	-	-
	unknown 20	1595	-	+	-	-
	humulene epoxide 2	1606	1606	+	-	-
	cubenol	1625	1642	tr	-	-
	alloaromadendrene oxide	1629	1643	+	-	-
	τ-cadinol	1637	1640	+	-	-
	α-cadinol	1649	1653	tr	-	-
	unknown 23	1651	-	+	-	-
	unknown 24	1655	-	+	-	-
	eudesma-4,11-dien-2-ol	1681	1690	tr	-	-
	unknown 26	1697	-	+	-	-
	cadina-4,10(15)-dien-one	1738	1755	tr	-	-
	unknown 27	1843	-	-	-	tr
diterpene	dimyrcene	1949	1958	tr	-	-
	abietatriene	2047	2054	tr	-	-
oxygenated diterpene	13-epi-manoyl oxide	1984	2014	tr	-	-
alkene	8-heptadecene	1674	1675	+	-	-
	5-nonadecene	1870	1873	tr	-	-
alkane	octadecane	1797	1800	tr	-	-
	nonadecane	1897	1900	tr	-	-
	eicosane	1998	2000	tr	-	-
	heneicosane	2094	2100	tr	-	-
	pentacosane	2493	2500	tr	-	-
other	allyl isothiocyanate	877	885	-	tr	-
unknown	unknown 1	750	-	-	+	-
	unknown 2	956	-	-	+	-
	unknown 6	1182	-	+	-	-
	unknown 7	1200	-	-	+	-
	unknown 9	1323	-	-	+	-
	unknown 12	1553	-	+	-	-
	unknown 13	1554	-	+	-	-
	unknown 15	1568	-	-	-	+
	unknown 21	1604	-	-	+	-
	unknown 22	1614	-	+	-	-
	unknown 25	1687	-	-	+	-

^a^ Mean RI were available from the NIST 2020 library. ^b^ Relative peak areas (TIC) were indicated as follows: “+++++”: 60% > *x* ≥ 40%; “++++”: 40% > *x* ≥ 20%; “+++”: 20% > *x* ≥ 1%; “++”: 1% > *x* ≥ 0.1%; “+”: 0.1% > *x* ≥ 0.01%; “tr”: *x* < 0.01%; “-“: not detected. ^c^ stereochemistry unknown.

## Data Availability

Data is contained with the article or Appendix A.

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
