# Peer review of "Characterization of Flavor Compounds in Distilled Spirits: Developing a Versatile Analytical Method Suitable for Micro-Distilleries"

_foods, 2022, doi:10.3390/foods11213358_

Round 1
Reviewer 1 Report
The article is large and written in details. There is no reference [1] at page 1 in the text. There are several questions, without answers to which the article cannot be published now.
The first question concerns the unknown compounds listed in Table 4 (more than 10 percent of total number of compounds). What are these compounds? Why can't they be defined? Either their existence should not be mentioned, or their analysis is needed.
Also in Table 4 Relative peak areas are indicated as "+++", "++", etc.
For example, "+++" corresponds to a value greater than 1%. Is it 1.1% or 10% or 50%?
I think, that it is necessary to give these values ​​in numbers instead of "+++", "++", etc. So that, if necessary, some analysis of the compounds can be carried out.
It is not clear from the article why many peaks should be detected if their values ​​are equal to "-" and "tr". And the main remark. The article is devoted to method for small businesses (micro-distillleries), as its title says:«...Analytical Method Suitable for Micro-Distilleries». The article does not define what is a micro-distilllery - the number of personnel, the volume of production, etc. But I am sure that no such «micro-distilllery» will be able to purchase the required equipment (GS-MS), maintain it in working order and perform daily analyzes. If you do not perform daily analyzes on very expensive equipment, then it does not make sense to buy it. In addition, such small business must have qualified personnel to maintain the equipment and perform tests. It is obvious that if necessary (and this will be only several times a year), they can send samples for analysis to large collective centers. And in such centers it makes no sense to talk about the complexity of analytical methods specifically for micro-enterprises. Therefore, the title should be changed. The mention of micro-distillleries should be removed also from the text and the focus should simply be on the analysis of methods. So, this paper needs major revision.
Reviewer 2 Report
This paper represents a classic example of a research in which the topic is a comparison of differnet extraction models. The results obtained with SPME method are, as expected, the best. Also pay attention on the space between rows (exp. Line 339).
Reviewer 3 Report
In the current manuscript, authors show a comparison of four different analytical methodologies (extraction techniques) for the identification of volatile and semivolatile compounds responsible for sensorial properties of spirits, using GC-MS as determination technique. The novelty of the four considered procedures is low, since all of them have been previously tested to cope with the kind of samples considered in the manuscript. However, the manuscript is very well organized and it provides valuable information regarding the benefits and limitations of each of the techniques, regarding the detectability of compounds from several chemical families and with different sensorial thresholds. Taking into account this latter comment:
1. Conclusions section. In the opinion of this reviewer, the 1st, 2nd and 4th paragraphs of this section are not conclusions derived from the study; thus, they have to be deleted.
2. Title and further comments in the manuscript. Authors recommend the HSSPME procedure to control the sensorial properties of spirits produced in micro-distilleries as compromise between efficiency, cost, sample amount requirements and a number of additional parameters. In my opinion, a factor to consider is whether these facilities have a GC-MS instrument dedicated to control their production or if analysis are performed by an external laboratory. In the second case, off-line SPE can be easily carried out in-situ and then sorbents, or sorbent extracts, can be send to laboratory for analysis. In case of HSSPME, fibers are relatively expensive; thus, samples (instead of extracts) have to be sent for analysis. Please consider this remark when revising the manuscript.
Round 2
Reviewer 1 Report
Some of my previous comments have been noted. The text has been amended accordingly.
But I cannot agree with the new phrase (p. 2) "Because such equipment is now becoming affordable, a GC/FID or a single quadrupole GC-MS being for instance in the range of 30,000 euros, some micro-distilleries have the ability to acquire such apparatus and carry out analysis of their spirits themselves".
No one will buy a device for 30,000 euros, which then will not be used effectively. To be effective such device should be used every day and many times per day. This is not the case of "small-scale", "micro-" distillery.
But I totally agree with the second sentence "They might also have access to such equipment through agreements with academic research laboratories, or subcontract the analyses. In every case, proposing a versatile and easy to implement method seems of interest and might encourage micro-distilleries to analyze their products."
So, in this case, the size of distillery ("micro-" or "macro-") is not important. A specilized laboratory can do any analysis for any client.
That is why it seems to me that the statements in this paragraph should be clarified once again.
And unknown components (Table 4) have not been investigated again. What it is? How do they affect the quality of the sample? Maybe it's just errors in the device detection? This should be clarified or excluded from consideration.
